# Exploring Competitive Relationship Between *Haemophilus parainfluenzae* and Mitis Streptococci via Co-Culture-Based Molecular Diagnosis and Metabolomic Assay

**DOI:** 10.3390/microorganisms13020279

**Published:** 2025-01-26

**Authors:** Yeseul Choi, Jinuk Jeong, Youngjong Han, Miyang Han, Byungsun Yu, Kyudong Han

**Affiliations:** 1Department of Microbiology, College of Bio-Convergence, Dankook University, Cheonan 31116, Republic of Korea; choiye402@gmail.com (Y.C.); hanyj0918@gmail.com (Y.H.); hanmyang227@gmail.com (M.H.); 2Smart Animal Bio Institute, Dankook University, Cheonan 31116, Republic of Korea; viem1273@gmail.com; 3Center for Bio-Medical Core Facility, Dankook University, Cheonan 31116, Republic of Korea; 4Department of Biomedical Sciences, College of Bio-Convergence, Dankook University, Cheonan 31116, Republic of Korea; ybs901287@gmail.com; 5Department of Human Microbiome Research HuNbiome Co., Ltd., R&D Center, Seoul 08507, Republic of Korea

**Keywords:** oral microbiome, dental caries, bacterial co-culture, metabolomics

## Abstract

Various bacterial strains with nitrate-reducing capacity (NRC), such as *Haemophilus*, *Actinomyces*, and *Neisseria*, are known to promote NH_3_ production, control pH in the oral cavity, and inhibit the growth of aciduric bacteria. However, experimental evidence on various estimated bacterial networks within the salivary microbiome is insufficient. This study aims to explore potential bacterial compositional competition observed within saliva samples from dental caries patients through a co-culture assay of mitis Streptococci, which is a primary colonizer in the salivary microbiome, and nitrate-reducing bacteria *Haemophilus parainfluenzae*. We investigated bacterial growth efficiency change by co-culture time using the qRT-PCR method. In addition, we applied LC/Q-TOF-based metabolites screening to confirm metabolic interactions between oral bacterial species and their association with dental caries from a metabolomics perspective. As a result, we first found that the nitrate reduction ability of *H. parainfluenzae* is maintained even in a co-culture environment with the mitis Streptococci group through a nitrate reduction test. However, nitrate reduction efficiency was hindered when compared with monoculture-based nitrate reduction test results. Next, we designed species-specific primers, and we confirmed by qRT-PCR that there is an obvious competitive relationship in growth efficiency between *H. parainfluenzae* and two mitis Streptococci (*S. australis* and *S. sanguinis*). Furthermore, although direct effects of nitrate reduction on competition have not been identified, we have potentially confirmed through LC/Q-TOF-based metabolite screening analysis that the interaction of various metabolic compounds synthesized from mitis Streptococci is driving inter-strain competition. In particular, we constructed a basic reference core-metabolites list to understand the metabolic network between each target bacterial species (*H. parainfluenzae* and mitis Streptococci) within the salivary microbiome, which still lacks accumulated research data. Ultimately, we suggest that our data have potential value to be referenced in further metagenomics and metabolomics-based studies related to oral health care.

## 1. Introduction

In the context of homeostasis balance between the various microbial compositional networks that inhabit each human body site, many microbiome-related studies suggest that preventing microbial dysbiosis is critical to maintaining human health [1,2]. Like the intestines and skin, the human oral cavity is a body site that maintains microbial-balanced homeostasis with at least 700 species of microorganisms and many still unknown species [3,4]. Based on this point, dentistry-related researchers are attempting to discover the relationship between microbial dysbiosis within the oral cavity and various oral diseases [5,6,7].

Recently, Jeong et al. reported the results of establishing a Korean-specific oral microbiome health control database for 112 Koreans without any oral disease and exploring the differences in bacterial compositional networks at the species level compared with 39 oral disease patients (dental caries and periodontitis [8,9]). They emphasized that the decreasing relative proportion of ‘nitrate-reducing bacteria (e.g., *Haemophilus parainfluenzae* and *Prevotella melaninogenica*)’ was commonly involved in each oral disease, and accordingly, it is related to the dominant phenomenon of primary colonizer bacterial species within the salivary microbiome that induces the expansion of each oral disease [10,11]. In particular, the results of their bacterial compositional network related to dental caries showed that the interaction between a substantial decrease in relative proportion for nitrate-reducing bacteria and the dominance of the ‘mitis Streptococci group (primary colonizer)’ within the salivary microbiome was observed in Korean data.

The nitrate-reducing bacteria are beneficial players within the human oral cavity by producing ammonium ion (NH_4_^+^) that control pH balance in the oral cavity and nitric oxide (NO) that controls host immune response and antibacterial effect in the process of reducing nitrate supplied into the oral cavity [12,13]. The mitis Streptococci group (e.g., *Streptococcus mitis*, *Streptococcus australis*, and *Streptococcus sanguinis*), like *Streptococcus salivarius*, is known as the dominant normal microflora with a relatively high-proportion within the human salivary microbiome composition [14]. However, recent studies have considered them an opportunistic pathogen group as a primary colonizer within saliva involved in dental caries progression, showing the two-sided characteristics of the mitis Streptococci group [15,16].

In this regard, there have been some reports of research cases investigating the microbial metabolic characteristics and disease association between the ‘mitis Streptococci group’ and ‘nitrate-reducing bacteria’ in the oral cavity [12,17], but clear microbiological findings on the mutual competition (e.g., growth and metabolic effect) within the salivary microbiome between the two are insufficient. Therefore, we aimed to demonstrate comprehensive experimentally the mutual competition relationship between the nitrate reduction bacteria ‘*Haemophilus parainfluenzae*’ and the ‘mitis Streptococci group (*Streptococcus mitis*, *Streptococcus australis*, and *Streptococcus sanguinis*)’ and the association with dental caries, as reported by Jeong et al. [9].

For this study, we performed a co-culture assay under various experimental conditions (culture method, nitrate treatment, and initial inoculum amount of *H. parainfluenzae* and mitis Streptococci) for each bacterial species. In this process, the growth competition relationship between mutual bacterial species was observed through qRT-PCR-based molecular diagnostic techniques. In addition, a nitrate reduction test was conducted to investigate the effect of the nitrate reduction activity of *H. parainfluenzae* on the competitive relationship with mitis Streptococci. In particular, LC/Q-TOF-based chemical analysis was performed to profile bacterial extracellular metabolites within the co-culture medium to complement the identification results for growth competition among each bacterial species from a metabolomics perspective. Ultimately, our research suggests that there is a potential value that can be used as reference data for microbiological etiology studies of dental caries by elaborating on the competition between *H. parainfluenzae* and mitis Streptococci group within the salivary microbiome.

## 2. Materials and Methods

### 2.1. Data Availability of Cited and Applied Public Oral Microbiome Data

The Korean healthy oral microbiome data for 112 subjects applied to the comparative microbiome analysis with the two oral disease groups recruited in the present study are registered in the National Center for Biotechnology Information (NCBI) reference database. The public annotation information is as follows: Viewing and downloading raw data on this public data are possible through the “PRJNA940351”. Accession link: https://www.ncbi.nlm.nih.gov/bioproject/PRJNA940351/ (accessed on 23 January 2025).

### 2.2. Preparation of Bacterial Strains and Optimizing of Cultivation Condition

Four standard bacterial strains (*Haemophilus parainfluenzae* KCTC 15417, *Streptococcus mitis* KCTC 13047, *Streptococcus australis* KCTC5657, and *Streptococcus sanguinis* KCTC 3284) for application for this study were airlifted from the KCTC (Korean Collection for Type Cultures, Jeollabuk-do, South Korea; Appendix A). After confirming the pure cell (no contamination of another bacterial species) of each bacterial species by pure culture method, the optimal seed culture of each bacterial strain was conducted by maintaining at 37 °C growth temperature and 5% CO_2_ air condition in a Heraeus^®^ BB15 incubator (ThermoFisher Scientific, Waltham, MA, USA). The basal culture medium composition used for all bacterial culture assays performed in the present study was equally optimized with the addition of 0.01 g NAD^+^ (Nicotinamide Adenine Dinucleotide) per 100 mL of BHI (Brain Heart Infusion, MB-B1008, MBcell, Seoul, South Korea) broth. NAD^+^ is an essential growth factor for *H. parainfluenzae*, but it has been confirmed that it does not affect the growth of all Streptococci applied to the present study [18]. Each seed-cultured bacterial species was suspended within a 20% glycerol cell stock solution and then stored at −80 °C for the after-procedure.

### 2.3. Bacterial Monoculture Assay and Growth Curve Analysis

Each bacterial species suspended in glycerol stock solution was inoculated by measuring an OD_600_ (Optical density value) = 0.3 (approximately 1 × 10^7^ CFU) via the NanoDrop One spectrophotometer (ThermoFisher Scientific, USA) to 150 mL of NAD^+^-added the BHI broth for bacterial monoculture assay. Monoculture analysis was conducted for 24 h, and the growth curve was determined by measuring the OD600 value every 2 h using a NanoDrop One spectrophotometer (ThermoFisher Scientific, USA) in triplicate for each bacterial species, and the average value was obtained.

### 2.4. Bacterial Co-Culture Assay

The description of the experimental comparison set for confirming the competitive relationship of each bacterial species via co-bacterial culture assay was as follows: The ‘set 1’ was inoculated 1:3 with *H. parainfluenzae* (H) and mitis Streptococci (S) group ratio, and the batch culture method was used. The ‘Set 2’ and ‘Set 3’ were inoculated with an H:S = 3:1 ratio, and batch and continuous culture methods were used, respectively (Table 1). All comparison sets included a case group (supplemented 0.757 g potassium nitrate; KNO_3_ positive) and a negative group (no supplemented potassium nitrate; KNO_3_ negative) for observing the nitrate reduction effect of *H. parainfluenzae* during co-culture assay [19]. The experimental methods for bacterial co-culture were as follows: For bacterial co-culture analysis, each glycerol-stocked bacterial cell was inoculated after normalizing bacterial load to an OD_600_ value of 0.3, and the *H. parainfluenzae* and mitis Streptococci ratios in each comparison set were considered (the total amount of inoculated liquid medium suspended for each bacterial species was 375 μL; [20]). The bacterial co-culture assays were conducted for 24 h in a 5% CO_2_, 37 °C culture environment with the LaboShker R100 equipment (LaboGene, Denmark), and 1 mL of culture medium was sampled triplicate every two hours for qRT-PCR-based bacterial load quantification. Additionally, pH values were checked every two hours using Orion™ Versa Star Pro™ Benchtop pH Meter (ThermoFisher Scientific, USA) during the co-culture period to confirm the change in acidity within the co-culture medium environment and the competitive relationship between each bacterial strain.

### 2.5. Nitrate Reduction Test

To confirm the nitrate reduction capacity of each bacterial strain, a nitrate reduction test was performed using a monoculture medium of each strain performed in an environment containing nitrate (Potassium nitrate; KNO_3_). Upon monoculture, each bacterial strain was inoculated into 0.202 g nitrate (final concentration: 50 Mm)-added 40 mL BHI broth at a dilution of OD_600_ = 0.3 (1 × 10^7^ CFU), and each culture medium collection to confirm the nitrate reduction reaction was performed after 24 h of incubation. Nitrate reduction tests were performed according to the experimental guidelines of the Sigma Aldrich Nitrate Reduction Test Kit (Sigma-Aldrich, St. Louis, MO, USA), and detailed methods applied in this study are as follows: To 2 mL of the collected culture, 100 uL of reagent A (sulfanilic acid) and reagent B (α-naphthylamine) were added, respectively, and the reduction reaction to nitrate was confirmed by observing that the medium turned red color. Additionally, when no reduction reaction was observed at the addition of each reagent solution, a trace amount of zinc dust was added to the culture medium to cross-check whether nitrate was saturated and whether it was reduced. This test method was applied equally to determine the effect of the nitrate reduction reaction of *H. parainfluenzae* on the growth of Streptococci within a co-culture environment.

### 2.6. Bacterial Species-Specific Primer Sets Design

To quantify bacterial load within bacterial culture media using the qRT-PCR method, CDS (Coding sequence region) information of the target bacterial genes (*H. parainfluenzae*: heme lyase NrfEFG subunit NrfF; *NrfF*, *S. mitis*: prephenate dehydratase; *pheA*, *S. australis*: prephenate dehydratase; *PheA*, *S. sanguinis*: nickel ABC transporter substrate-binding protein; *NikA*) was obtained from the National Center for Biotechnology Information Database (NCBI DB) for species-specific primer sets design (Appendix A). Each bacterial target protein was selected using the ‘OrthoVenn3’ web-based tool, which allows the exploration of distinct genes specifically possessed by the whole-genome comparative alignment analysis between each bacterial species. It was difficult to find specific genes to target each bacterial species separately in the case of *S. mitis* and *S. australis* because they were linked by very close branches on the phylogenetic tree. Therefore, a specific region capable of targeting each of them within the same gene with a large number of sequence variation sites was selected for primer design. The selection of specific primer regions for targeting each bacterial species was performed by the Multiple Sequence Alignment (MSA) method using the BioEdit 7.2v software, which can identify the nucleotide sequence variable region between each bacterial species. In the MSA method, genetic sequence information on multiple strains (=strain level) of the same species known to have been isolated from the human oral cavity was applied as input values to improve primer target accuracy for each salivary bacterial species (Appendix A). Experimental suitability (Tm value, GC%, and potential for primer-dimer to form, etc.) of the selected primer set was confirmed through the in silico test using the Oligo calc and Oligo Analysis open web tool (https://www.bioinformatics.org/JaMBW/3/1/9/index.html, accessed on 17 March 2024; https://eurofinsgenomics.eu/en/ecom/tools/oligo-analysis/, accessed on 17 March 2024). Finally, the species-specific accuracy for the amplicon sequence region covered by each primer set was verified by the NCBI Nucleotide BLAST in silico test and PCR amplification (Appendix A).

### 2.7. Quantification of Relative Bacterial Load Using qRT-PCR

#### 2.7.1. Bacterial Genomic DNA Extraction

A total of 20 (ten negative groups and ten case groups) 1 mL culture medium samples within 1.5 mL conical sampling tubes per each comparison set were centrifuged at 4200 rpm for five minutes to form bacterial cell pellets. The supernatant was removed, and bacterial genomic DNA (gDNA) was extracted from the pellet using the QIAamp PowerFecal Pro DNA Kit (QIAGEN, Hilden, Germany). This process was performed for triplicate samples.

#### 2.7.2. Bacterial Quantification Using qRT-PCR Method

The equipment and reagents used in the experiment were the StepOnePlus™ Real-Time PCR (ThermoFisher Scientific, USA) at the Center for Biomedical Engineering Core Facility (Dankook University, South Korea) and the Quanti-Speed SYBR No-Rox kit (PhileKorea, Seoul, South Korea). Each sample’s concentration was equally normalized to 10 ng/ul through dilution, a process that was then confirmed using the Qbit Fluorometer 4.0 v and the 1X dsDNA HS Assay kit (ThermoFisher Scientific, USA). The qRT-PCR method-based normalization concentration analysis applied with a primer pair targeting the bacterial 16 S V5 hyper-variable region was performed to normalize the potential bacterial gDNA density of each sample. The sequence information of the bacterial 16 S V5 region-specific universal primer pair is as follows: Forward primer (785F): 5′-GGATTAGATACCCTGGTA-3′, Reverse primer (907R): 5′-CCGTCAATTCMTTTRAGTTT-3′. After setting the annealing temperature (*S. mitis* and *S. australis*: 60.5 °C, *S. sanguinis* and *H. parainfluenzae*: 61 °C) with the working test of the designed species-specific primer set, Ct-value data for each strain per sample were obtained by qRT-PCR (Appendix A). This process was performed for triplicate samples, and the average value was calculated.

### 2.8. Extracellular Metabolite Screening Using LC/Q-TOF Method

For analysis, culture medium samples, which were collected from three sampling points (‘Time 1’, ‘Time 4’, and ‘Time 10’) per comparison sets, were selected according to qRT-PCR data, and additional monoculture-based culture medium samples with or without the addition of KNO_3_ was analyzed for comparison. All 10 mL of microbial culture samples were centrifuged twice (4200 rpm, 5 min) to obtain supernatants. The supernatant was freeze-dried before LC/Q-TOF MS/MS analysis and redissolved in MeOH through a pretreatment process. For the reliability of the assay, all samples were prepared in triplicate. The LC/Q-TOF MS/MS analysis was performed using a 1290 Infinity II Series LC system and an Agilent 6546 series Q-TOF mass spectrometer at the Center for Biomedical Engineering Core Facility (Dankook University, South Korea). Analysis was performed on both polar and non-polar columns. The non-polar column samples were separated using a ZORBAX Eclipse Plus C18 Rapid Resolution HD (2.1 × 150 mm, 1.8 micron) at a flow rate of 0.3 mL/min. The polar column samples were separated using a Proshell 120 HILIC-Z (2.1 × 100 mm, 1.9 um) at a flow rate of 0.3 mL/min. The mobile phase was water containing 0.1% formic acid (A) and ACN containing 0.1% formic acid (B), and the following gradient was used: 0–30 min 5% B, 30–40 min 95–98% B and 40–45 min 5%. The mass scan range was 50–1100 *m*/*z*. The Mass Profiler Professional (MPP) tool was used to compare metabolites over time within the co-culture comparison set. MPP can be used in any MS-based differential analysis containing two or more sample groups or variables [21]. It has an ID browser, statistical analysis tools, and visualization tools (heatmap, PcoA plot). We also used MPP’s ‘Filter by frequency tool’ to obtain a merged compound list of the 3 replicate samples. For MPP analysis, MassHunter Profinder software 10.0 (Profinder) was used for extraction and data review, filtering data with a minimum quality score of 90. Also, MassHunter Qualitative analysis software 10.0 (MassHunter Qual) was used to confirm target or suspect compounds and identify unknown analytes.

### 2.9. Statistical Analysis

All data were analyzed using the GLM procedure (SAS Inst. Inc., Cary, NC, USA). Data variability was expressed as standard error. The *p*-value < 0.05 was considered statistically significant. The Kruskal–Wallis and Mann–Whitney statistical tests were used to compare the metabolite screening analysis data to identify significant metabolic compositional differences in each comparison group. This non-parametric statistical analysis was performed using GraphPad PRISM v8 (GraphPad Software Inc., La Jolla, CA, USA). The Permutational multivariate analysis of variance (PERMANOVA) non-parametric statistical test was used to confirm the statistical significance of distance dissimilarity (PCoA analysis) between each comparison group.

## 3. Results

### 3.1. Summary of Previous Study and Workflow for Present Study

In a previous study by Jeong et al. [9], they investigated the bacterial compositional network by comparing the oral microbial diversity and composition of 26 dental caries patients and 112 healthy oral Korean subjects (KOGA type). They confirmed through NGS-based microbiome analysis that the relative frequency of nitrate-reducing bacteria ‘*Haemophilus parainfluenzae*’ was lower in the dental caries group than in the healthy group, while the relative frequency of bacterial species corresponding to ’*Streptococcus* unclassification’ was higher than the healthy group (Showing Figures 4 and 5 in [9]). In addition, this study confirmed through comprehensive microbial molecular diagnostics that several bacterial species belonging to the ’Mitis Streptococci group’ inlayer the category classified as ’*Streptococcus* unclassification’ (Showing Appendix A and Figure 4 in [9]). Therefore, based on their results, we designed a microbial experiment (monoculture and co-culture assay)-based workflow using qRT-PCR-based molecular diagnostics and LC/Q-TOF-based bacterial metabolic network profiling to comprehensively verify the potential mutually competitive relationship (growth effect, metabolic network, and effect of nitrate reduction) within the human oral cavity of nitrate-reducing bacteria *H. parainfluenzae* and three mitis Streptococci group bacterial species (*Streptococcus mitis*, *Streptococcus australis*, and *Streptococcus sanguinis*; Figure 1).

### 3.2. Optimization of Co-Culture Conditions and Growth Curve Assay

Prior to conducting an experimental process for this study, *H. parainfluenzae* utilizes nicotinamide adenine dinucleotide (NAD^+^; factor V) as an essential growth factor during cell growth, so we first needed to determine whether this affects the growth of the mitis Streptococci group within the co-cultured environment with *H. parainfluenzae* [22]. Therefore, we validated this consideration point through a monoculture-based growth curve assay for each bacterial species for 24 h within BHI broth with NAD^+^ (Figure 2; Appendix A). As a result, the growth curve confirmed that growth gradually increased over time with all species equally. In addition, it was commonly confirmed that all bacterial species entered the exponential phase six hours after cultivation. With this result, we demonstrated that NAD^+^ had no effect on the growth efficiency for mitis Streptococci and confirmed that the cell growth rate of all bacterial strains was similar.

### 3.3. Nitrate Reduction Assay on Bacterial Co-Culture Environment

First, we identified the nitrate-reducing capacity for each bacterial species under monoculture with nitrate-treated conditions, respectively. As a result, we verified that only *H. parainfluenza* reduced nitrate to nitrite through the results of the ‘Reagent A (sulfanilic acid)’ treatment reaction (Appendix A). Additionally, when a small amount of zinc dust treatment reaction was used to confirm the presence of residual nitrate within each bacterial culture medium, the color change was not detected within the culture medium of *H. parainfluenzae* (no presence of residual nitrate). However, it was confirmed that nitrate was contained within the culture medium of all species except *H. parainfluenzae* (the culture medium color was changed to red). We cross-checked for these results that only *H. parainfluenzae* possessed nitrate-reducing capacity, and it reduced all of the treated nitrates within the culture medium during the 24 h cultivation time. Based on these results, we performed a nitrate reduction test to determine the association between reducing the nitrate of *H. parainfluenzae* and the potential mutual competition among bacterial species within the co-culture environment. The bacterial co-culture assay was conducted using three comparison sets (Table 1, detailed in the Section 2). As a result, we confirmed that the change in the color of the culture medium (accumulation of nitrite; nitrate reduction positive reaction) started to appear after 10 h point of cultivation time for all comparison sets (Figure 3). In particular, we found that the accumulation rate of nitrite was higher in sets 2 and 3, where the initial inoculation amount of *H. parainfluenzae* was high through color-changing levels for each culture medium. Finally, when a small amount of zinc dust was treated to determine the presence of residual nitrate within all culture mediums, we observed strong color changes to red from all comparison sets (nitrate detection positive). From this result, we found that the nitrate reduction efficiency of *H. parainfluenzae* decreased within the co-culture environment, in contrast to the monoculture results. In this respect, we checked that the nitrate-reducing capacity of *H. parainfluenzae* becomes active from the middle point of the co-culture with or without the mitis Streptococci group. Still, we deduced that the nitrate reduction efficiency was affected by potential interaction factors (e.g., metabolism-associated with inhibition of nitrate reduction capacity or inhibition of growth for *H. parainfluenzae*) with the mitis Streptococci group.

### 3.4. Investigation of Potential Growth Effect Between Each Bacterial Species

We performed a bacterial co-culture assay based on optimized culture conditions for all bacterial species to identify potential growth inter-relationships within the salivary microbiome. The bacterial co-culture assay was conducted by three comparison sets (Table 1; detailed in Section 2), and we determined the change in pH and relative bacterial load within the culture medium according to incubation time (Figure 4; Appendix A). In addition, we grafted the results of the nitrate reduction assay, which we previously identified, to determine the association.

#### 3.4.1. Change In Ph Value Within Culture Medium

When we observed changes in pH within the co-culture medium (Appendix A), we confirmed that the initial pH value (sampling points ‘Time 1’ to ‘Time 2’) of set 1 was approximately 6.9, with relatively low values calculated relative to sets 2 (approximately 7.2) and 3 (approximately 7.3). In addition, we could confirm in common from all comparison sets that the pH value proceeded to decrease sharply at the point of incubation after ‘Time 2–3’ and from ’Time 6’ and checking an acidic environment with a pH value of about 6.0 or less was maintained within the culture medium. From this result, considering that the initial pH value (between sampling points ‘Time 1–3’) within the culture medium of set 1 was lower than that of sets 2 and 3, we deduced that the initial *Streptococcus* dominance condition would have provided a potential factor (e.g., acidic-related metabolic pathway) the change in pH environment within the medium. Contrary to our supposition that ammonium ions produced by nitrate reduction in *H. parainfluenzae* would affect pH control within the medium, the addition of nitrate did not affect the pH change pattern.

#### 3.4.2. Observation of Relative Bacterial Load Changes Based on qRT-PCR Method

Each bacterial genomic DNA (gDNA) sample, obtained from a co-cultured medium according to cultivation time, was standardized to the same concentration (10 ng/uL) to quantify the relative proportion change over time for each bacterial species. When we observed the growth rate of qRT-PCR-based relative bacterial load changes using targeted-specific primer sets for each bacterial species, we found from all comparison sets that the growth rate of *H. parainfluenzae* at sampling points ‘Time 1–3’ was higher than that of two mitis Streptococci group except *S. mitis*. Considering the results of the growth curve analysis (Figure 2), we could confirm that the growth efficiency *of H. parainfluenzae* increased within the initial co-culture environment, regardless of the difference in inoculating amount between *Haemophilus* and *Streptococcus* (H:S ratio) and affected the growth lag of *S. australis* and *S. sanguinis*. In addition, we confirmed from our results that there was no growth competition between *S. mitis* and *H. parainfluenzae*. This result implies that *S. mitis* is likely excluded from the candidate species of the ‘*Streptococcus* unclassification’ group, which showed a negative relative frequency correlation with *H. parainfluenzae* predicted by Jeong et al.’s research [9]. Contrary to these results for initial co-culture time, we identified that *S. australis* and *S. sanguinis* had higher growth rates than *H. parainfluenzae* in the mid to late co-culture period (sampling points ‘Time 6–10’) in all comparison sets. In addition, a stagnation pattern with no change in the growth rate of *H. parainfluenzae* was observed within this cultivation time. With these results, we demonstrated that the mutual competition (negative correlation) of growth between *H. parainfluenzae* and the two mitis Streptococci was formed. In addition, we also confirmed that sets 2 and 3, which had high initial inoculation doses of *H. parainfluenzae*, reached earlier the point with the highest growth efficiency of *H. parainfluenzae* compared to set 1, but that they had no effect on the inhibition of growth of two mitis Streptococci in the second half of cultivation.

When examining the mutual growth competition among bacterial species according to nitrate treatment, as with the result of observations of pH environment changes within the culture medium, we found no distinct differences in the qRT-PCR results between the negative control and nitrate treatment groups of all comparison sets. Therefore, we predicted that the nitrate-reducing capacity of *H. parainfluenzae* was not the main factor influencing the mutual growth competition relationship among bacterial species within the co-culture environment and that there would be other external factors (e.g., bacterial metabolism other than nitrate reduction).

#### 3.4.3. Association Between Relative Bacterial Frequency Change and pH Condition

We investigated the association between changes in the pH environment within the culture medium according to co-culture time and changes in qRT-PCR-based relative bacterial loads. In comparing each experimental condition (comparison sets), the pH value of set 1 with a high initial inoculation rate of *Streptococcus* was generally measured to be lower than that of sets 2 and 3. Furthermore, at the mid-to-late time point of co-culture (‘Time 6–10’) with high growth rates of *S. australis* and *S. sanguinis*, an acidic environment of pH 6.0 or lower was observed in all comparison sets. Conversely, a pH environment of about 7.0 was maintained at the initial time point of cultivation (normal pH condition of healthy human oral cavity; [23,24]), when the growth rate of *H. parainfluenzae* was high in all comparison sets, at which time the change in growth rates of *S. australis* and *S. sanguinis* stagnated. In this respect, we confirmed that it is a positive result for our prediction, in which two mitis Streptococci provided some factors (e.g., acid-related metabolism) in lowering the pH environment within the culture medium during co-culture assay. Additionally, we also deduced that the growth delay of *S. australis* and *S. sanguinis*, coupled with the pH decrease, was potentially related to the growth efficiency of *H. parainfluenzae* increased.

### 3.5. Metabolite Screening for Exploring Competition Relationship Between Bacterial Species

When comparing the qRT-PCR method-based relative bacterial growth efficiency, we found that there was an obvious mutual competition relationship between *H. parainfluenzae* and the two mitis Streptococci (*S. australis* and *S. sanguinis*). In the case of *S. mitis*, the competition relationship with *H. parainfluenzae* has not been identified, so the data were excluded. However, contrary to our predictions, this relationship was not significantly affected by the nitrate reduction in *H. parainfluenzae*, the difference in the initial inoculum amount by *Haemophilus* and *Streptococcus* (H:S initial ratio), and the bacterial cultivation method. Therefore, we applied LC/Q-TOF analysis-based metabolite screening to investigate potential factors that are likely to affect the competitive relationship among bacterial species from a metabolomic perspective and predicted an association between the metabolic effect and dysbiosis of *H. parainfluenzae* and two mitis Streptococci (*S. australis* and *S. sanguinis*) within the salivary microbiome.

#### 3.5.1. Metabolite Library Construction Based on Monoculture for Each Bacterial Species

We first performed metabolite screening on monoculture medium samples to confirm differences in metabolite composition among bacterial species (*H. parainfluenzae*, *S. asutralis*, and *S. sanguinis*) and constructed metabolite library sets that could reflect particular species-derived metabolites within the co-culture environment. In the negative group (non-treated nitrate), the average count of raw metabolic compounds (matching score > 90 using MassHunter Profinder software tool) screened from each bacterial species was 1142, and the average count of metabolic compounds that matched the reference database (Identification and annotated list of compounds using Agilent MPP software’s ID browser tool) was 363 (Table 2; Appendix A). For the case group (nitrate treatment), the average count of raw metabolic compounds was 1027, and for matched compounds in the reference database, it was an average of 310. The contrast of metabolic compositions between each bacterial species resulted in the number of distinct metabolic compounds of *H. parainfluenzae* being much higher than two mitis Streptococci. In the case of Streptococci, the number of metabolites shared between the two species was significantly higher than that of distinct metabolites per each species. We deduced that these results may be due to the two mitis Streptococci being linked to each other in a closely phylogenetic branch (allied species) and thus share a significant number of metabolisms [16,25].

Next, we confirmed by PCoA (Principal Coordinate Analysis) distance dissimilarity test that there were statistically significant differences in metabolic compound composition between the screened metabolite library sets from the monoculture medium of each bacterial species (Kruskal–Wallis test *p* < 0.05: *H. parainfluenzae* vs. two mitis Streptococci; Mann–Whitney U test = no significant: among Streptococci). We demonstrated an obvious difference in metabolite composition between *H. parainfluenzae* and mitis Streptococci through this result, and we also confirmed that the monoculture-based metabolite library sets we constructed could be a backbone database that could determine which bacteria the metabolites detected upon co-culture environment were from (Figure 5).

#### 3.5.2. Investigation of Metabolic Effect Among Bacterial Species Within Co-Culture Environment

We investigated the association for metabolic effect between bacterial species within the co-culture environment in connection with the mutual competitive relationship among species previously identified by qRT-PCR-based molecular diagnosis analysis. For this analysis, we focused on differences in the composition and relative proportions of metabolic compounds detected at that time point, as qRT-PCR results identified obvious differences in the growth competition among bacterial species between ‘Time 1’, ‘Time 4’, and ‘Time 10’ sampling points. This analysis was performed by metabolite screening shared between each cultivation time (inter-sampling points) and profiling the distinct metabolites for each cultivation time (intra-sampling points). In addition, we conducted profiling by focusing on three potential metabolic factors directly affecting a competition relationship among bacterial species: antibacterial effects, lipid metabolism, and pH condition regulation [26,27].

##### Shared-Metabolites Screening for Inter-Sampling Points

We filtered metabolic compounds, which were calculated log_2_ fold change in >2 (*p*-value < 0.05) from MPP analysis among the compounds shared at three sampling points (‘Time 1’, ‘Time 4’, and ‘Time 10’) for each comparison set. Then, we observed the relative proportion change about sorted metabolic compounds between each sampling point (Table 3 and Appendix A). In this MPP analysis, we found four patterns for the relative proportion change in shared-metabolic compounds between each sampling point in the co-culture environment, and we discussed them by associating them with growth efficiency investigation results based on the qRT-PCR method (Appendix A). The metabolites corresponding to ‘Pattern 1’ were progressively up-regulated from ‘Time 1’, ‘Time 4’, and ‘Time 10’, while pattern 2 was down-regulated. The metabolites corresponding to ‘Pattern 3’ show a regulation that rises and falls based on ‘Time 4’, and ‘Pattern 4’ was the opposite.

As a result, we confirmed that the relative proportion change between each sampling point was detected for a larger number of shared-metabolic compounds in set 1 compared to set 2 and set 3 (both Negative and Case groups). Considering that in the culture condition of comparison set 1, the initial inoculation ratio of Streptococci was higher than set 2 and set 3, we could find that the initial dominance rate of *Streptococcus* affected various bacterial metabolism by metabolic compounds within the culture medium during the entire co-culture period. In this analysis, we confirmed that in the comparison set 1, the proportion of metabolic compounds related to the lipid pathway and antibacterial effects was the highest in both the negative group and the case group (detail of shared-metabolic compounds information is indicated in Appendix A). Interestingly, unlike other comparison sets, we noted that most of the shared-metabolic compounds detected in set 1 corresponded to the ‘Pattern 4’ type. In addition, we confirmed in set 1 that some acid-related metabolic compounds associated with pH control were also detected. It reflects that our results correlated with the data that the pH value of set 1 was relatively lower than that of set 2 and set 3 in our observation of the pH change according to the co-culture time. We confirmed that most metabolic compounds related to pH regulation detected in set 1 also correspond to the ‘Pattern 4’ type (Appendix A). Considering that the relative proportion changing shape of metabolites shown in ‘Pattern 4’ is consistent with the type of change in growth efficiency of *Streptococcus* identified by qRT-PCR results, we deduced that a number of shared-metabolites corresponding to ‘Pattern 4’ detected in set 1, where the initial inoculation ratio of *Streptococcus* is high, are *Streptococcus*-derived compounds involved in the growth of *Streptococcus* and competition with *H. parainfluenzae*. In other words, we could predict that *Streptococcus*-synthesized metabolites produced for competition for growth with *H. parainfluenzae* early phase in culture had a lasting effect throughout the entire co-culture period. Additionally, like the nitrate reduction test results in this study, metabolic compounds involved in bacterial nitrate-reduction metabolism were not detected in both negative and case groups.

##### Distinct Metabolites Screening for Each Intra-Sampling Point

Next, we screened distinct metabolic compositions for each intra-sampling point to identify time point-specific metabolites detected when the competition between bacterial species was obvious Table 3 and Appendix A). Since this approach was not to investigate the metabolic compounds shared among three sampling points, it is not possible to calculate the relative proportion changes in the detected metabolites, but by identifying the time point-specific metabolites detected at each sampling point, we focused on investigating a list of metabolic compounds related to the three factors (Lipid pathway, pH regulation, and Antibacterial effect; detail of distinct metabolic compounds information is indicated in Appendix A) affecting the competition relationship between bacterial species screened in this study.

In this screening results, unlike the shared-metabolite analysis results, we confirmed that the number of distinct metabolic compounds was evenly detected for each sampling point in all comparison sets (detection count of average distinct metabolic compound = 23). However, on filtered data for three factors associated with bacterial competition, we observed that the number of distinct metabolic compounds related to the antibacterial effect was detected at a high rate within each sampling point in all comparison sets. This result potentially reflects that, regardless of bacterial culture conditions, various antibacterial effect-related metabolisms (*Streptococcus* or *Haemophilus* derived) interacted for inter-bacterial species growth and dominance competition according to each cultivation time point within co-culture environments. On the other hand, we found that the proportion of distinct metabolic compounds associated with pH regulation metabolism was the highest in set 1, which was the same as the shared-metabolite screening results. In particular, we noted that a higher number of acid-related metabolic compounds were detected at sampling point ‘Time 4’, where the growth efficiency of *Streptococcus* began to increase rapidly. This result reflects that the detection of acid-associated metabolites involved in acidification within the co-culture medium was also affected by an increase in growth efficiency, along with the initial inoculation ratio of *Streptococcus*. Furthermore, we identified that many distinct metabolic compounds related to lipid pathways other than pH-regulated metabolism were detected after the sampling point ‘Time 4’ in each comparison set. In particular, in the case of oral Streptococci, it is known to contribute to acidification by releasing large amounts of CO_2_ and fatty acids during glycolysis or lipid metabolism, lowering the pH within the growth environment [28]. In this respect, we could predict that the rapid decrease in growth efficiency of *H. parainfluenzae* after sampling point ‘Time 4’ involved several metabolisms associated with the lipid pathway and pH regulation of mitis Streptococci. In other words, it suggests that the abnormal dominance of two mitis Streptococci within the human salivary microbiome is likely to act as a major cause of pH decreasing within the oral cavity and dental caries. Additionally, like the nitrate reduction test results in this study, metabolic compounds involved in bacterial nitrate-reduction metabolism were not detected in both negative and case groups.

##### Potential Core-Metabolites Screening for Matching Monoculture-Based Metabolites Library

Finally, we filtered potential core-metabolic compounds matching the previously constructed monoculture-based metabolite library sets for each bacterial species on the list of compounds for each factor (Antibacterial effect, Lipid pathway, and pH regulation) affecting competition for growth efficiency among species listed in Table 3, and we discussed the potential causes of dysbiosis between *H. parainfluenzae* and mitis Streptococci in the salivary microbiome from a metabolomics perspective (Table 4). As a result, it was found that the number of compounds (total number of both Negative and Case groups) matched with mitis Streptococci (both *S. australis* and *S. sanguinis*)-derived metabolite library was significantly higher than that of *H. parainfluenzae* (Antibacterial effect: Streptococci = 11, *H. parainfluenzae* = 4; Lipid pathway: Streptococci = 4, *H. parainfluenzae* = 2; pH regulation: Streptococci = 2, *H. parainfluenzae* = 0). Based on these results, we determined that it could refer to understanding the mechanisms of intra-oral growth competition for *H. parainfluenzae*, taking into account the characteristics of the potential core metabolites of mitis Streptococci. For example, the ‘Minocycline’ is known as a tetracycline class of antibiotics that binds to the 30 S ribosome, interferes with bacterial mRNA decoding, and inhibits protein synthesis [29]. The minocycline is a mitis Streptococci metabolite library-matched compound detected at sampling point ‘Time 10’ in comparison set 1. This is consistent with our prediction that it may be a mitis Streptococci-derived compound found during the cultivation time when the growth efficiency of the two mitis Streptococci is at its highest, as a qRT-PCR result. ‘Traumatic acid’ is known to be a metabolic compound that can acidify pH conditions. A previous study has shown that the traumatic acid is also involved in reducing oxidative stress (antioxidant activity) within the environment [30]. Various species belonging to the oral bacteria produce H_2_O_2_ during growth activity, which is a cytotoxin that induces the death of phagocytes such as macrophages, neutrophils, and epithelial cells, causing oxidative stress within the oral cavity [31,32]. However, oxidative stress such as H_2_O_2_ can also inhibit the growth of bacteria, so it could be predicted that mitis Streptococci produces the traumatic acid as a metabolite to defend against the oxidative stress by *H. parainfluenzae* within the co-culture environment [33]. In this respect, understanding the metabolic characteristics of potential core-metabolic compounds, which screened from this study, may suggest the possibility as a potential clue to investigating the competition mechanisms of mitis Streptococci and *H. parainfluenzae* within the human salivary microbiome.

## 4. Discussion

In summary, we investigated the mutual competition relationship of *Haemophilus parainfluenzae* and mitis Streptococci (*S. mitis*, *S. australis*, and *S. sanguinis*) found from differences in bacterial networks within the salivary microbiome between healthy and dental caries patients for Korean by *Jeong* et al.’s previous study through various experimental approach based on bacterial co-culture assay [9].

We established optimal co-culture conditions for all bacterial species and, based on this, explored potential factors (initial compositional dominance ratio, pH, nitrate reduction, and metabolic effect) that could affect the inter-species competition. First, we verified the nitrate reduction activity of *H. parainfluenzae* through the nitrate reduction test (all mitis Streptococci have no capacity), and it was confirmed that nitrite was continuously detected from 10 h of cultivation during co-culture assay. However, contrary to the results of the monoculture-based nitrate reduction test, it was possible to predict the influence of mitis Streptococci on the change in the nitrate reduction efficiency of *H. parainfluenzae*, considering the strong detection of residual nitrate within the culture medium of the co-culture environment.

Next, we demonstrated that the mutual competition relationship between *H. parainfluenzae* and two mitis Streptococci (*S. australis* and *S. sanguinis*) was evident by identifying changes in the growth efficiency of each strain within the incubation time using qRT-PCR-based microbial molecular diagnostic methods. In this analysis, we confirmed that the pH environment within the culture medium at the time when the growth efficiency of both Streptococci predominated (from ‘Time 6’) for all comparison sets was acidified, which was associated with a factor that lowered the growth efficiency of *H. parainfluenzae.* The pH condition within the healthy human oral cavity is 6.7–7.5, and when the pH is imbalanced, diseases related to demineralization for tooth or supragingival sites can occur [34]. In this regard, changes in the pH condition due to the competition between *H. parainfluenzae* and mitis Streptococci are potential factors that may impair oral health, suggesting that it is important to maintain the balance of their proportion in the oral cavity. Considering that the nitrate-reducing activity of *H. parainfluenzae* was observed even when the growth efficiency of mitis Streptococci prevailed, we found that the nitrate-reduction activity of *H. parainfluenzae* did not directly affect the change in the growth efficiency of mitis Streptococci. However, besides *H. parainfluenzae*, various bacterial strains with nitrate reduction capacity reside within the human oral cavity. Since this bacterial group positively affects oral health, such as antibacterial effects and pH balance control within the oral cavity on acidogenic bacteria such as *Streptococcus mutans* [13,35], we suggest that continuous research to maintain the balance of nitrate-reducing bacteria within the oral cavity is essential.

Finally, when the LC/Q-TOF-based metabolite screening method was used to investigate the association between the bacterial growth competition relationship and the metabolic effects, we confirmed that a number of metabolites related to antibacterial effect and lipid metabolism were detected from each sampling point (‘Time 1’, ‘Time 4’, and ‘Time 10’) when there were distinct differences in growth efficiency among bacterial species. Considering that the antibacterial effect and lipid metabolism are major factors that directly affect bacterial growth and mutual competition, we noted that a number of these compounds were detected at intersection points where competition for growth efficiency among each bacterial species was evident [26,27]. Notably, many metabolic compounds shared between each sampling point were identified as the same type of relative proportion change (indicated ‘Pattern 4’ in our results) as the change in growth efficiency of two mitis Streptococci (*S. australis* and *S. sanguinis*). These results suggest that metabolisms of mitis Streptococci were more predominant in the competition among bacterial species within the co-culture environment. Additionally, we also confirmed that some pH regulation-related metabolic compounds that can cause acidification of pH conditions within the culture medium were detected. Metabolic compounds (acid or lipid pathway-related compounds) associated with pH regulation were detected at a high rate, especially in set 1, where the initial inoculation rate of mitis Streptococci was high. In addition, it accounted for a high proportion in the latter half of the co-culture, where the growth efficiency of mitis Streptococci was rapidly increased. In this regard, we could confirm that the pH-regulated metabolism of mitis Streptococci was a major factor in its competitive advantage over *H. parainfluenzae* within the co-culture environment. Furthermore, we predicted that a nitrate reduction reaction of *H. parainfluenzae* was detected late in the culture but did not significantly affect pH control due to the high dominance rate of mitis Streptococci and the resulting acidification-related metabolic activity. In addition, matching detected metabolites in the co-culture environment with the monoculture-based metabolite library for each bacterial species found that the proportion of compounds matching streptococci-derived metabolites was higher than that of *H. parainfluenzae*. We showed an approach for understanding competition between mitis Streptococci and *H. parainfluenzae* within the salivary microbiome and applicability as reference data by providing a list of these potential strain-derived core metabolites.

## 5. Conclusions

In conclusion, we explained the mutual competition relationship between *H. parainfluenzae* and mitis Streptococci through various experimental approaches. First, we verified by qRT-PCR-based molecular diagnostics that two of the mitis Streptococci group (*S. australis* and *S. sanguinis*), which predicted that there would be a mutual competition with *H. parainfluenzae* in previous studies, had a negative correlation relationship for growth efficiency change. However, contrary to our logical point, we confirmed that the nitrate-reducing capacity of *H. parainfluenzae* is not applied as the primary cause of the competitive relationship between bacterial species. Therefore, we explored the competitive relationship between bacterial species through extracellular metabolite screening methods other than nitrate reduction metabolism by applying LC/Q-TOF-based metabolite screening methods. In particular, we constructed basic reference data to understand the metabolic network between each target bacterial species (*H. parainfluenzae* and mitis Streptococci) within the human salivary microbiome, which still lacks the accumulated research data. Ultimately, our approach and results demonstrate that it is important to maintain a balance between mitis Streptococci and *H. parainfluenzae* within the human salivary microbiome. We also suggest that our data have potential value to be referenced in further metagenomics and metabolomics-based studies related to oral health care.

## Figures and Tables

**Figure 1 microorganisms-13-00279-f001:**
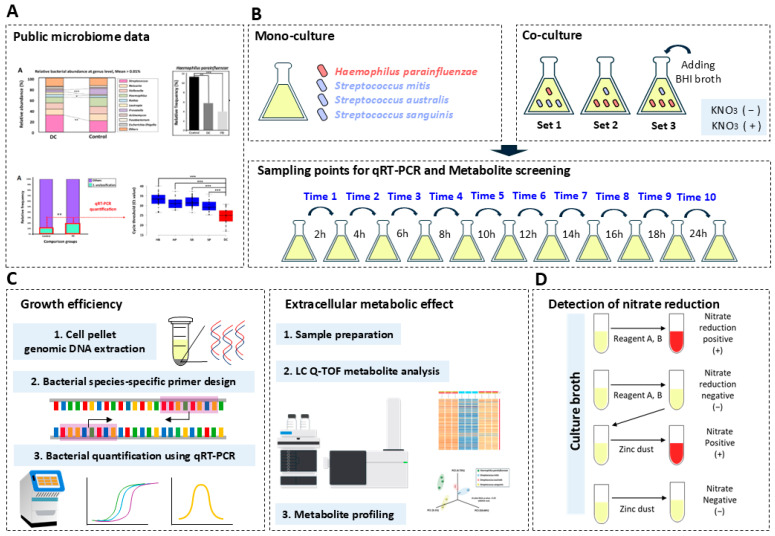
Schematic diagram of overall experimental workflow performed in the present study. The schematic diagram shows the overall experimental workflow carried out in this study to investigate the mutual competition relationship between *Haemophilus parainfluenza* and mitis Streptococci. (**A**) Citation to public microbiome data, which explained the negative compositional correlation between *H. parainfluenzae* and *Streptococcus* sp. (indicated in Figures 3–5 from Jeong et al. ’s previous study [9]). (**B**) Illustration showing the bacterial culture method (mono and co-culture) and sampling process for qRT-PCR assay and metabolite screening applied to this study. (**C**) The experimental process to demonstrate the two major factors for the competitive relationship between bacterial species focused on this study. (**D**) The process of nitrate reduction test and step-by-step result analysis method applied in this study.

**Figure 2 microorganisms-13-00279-f002:**
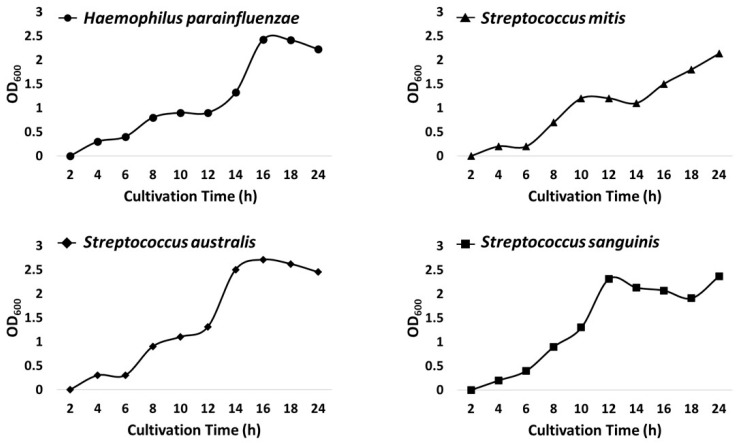
Growth curve graph for each bacterial species. These line graphs show the calculated growth curve analysis data when performing monoculture for each bacterial species.

**Figure 3 microorganisms-13-00279-f003:**
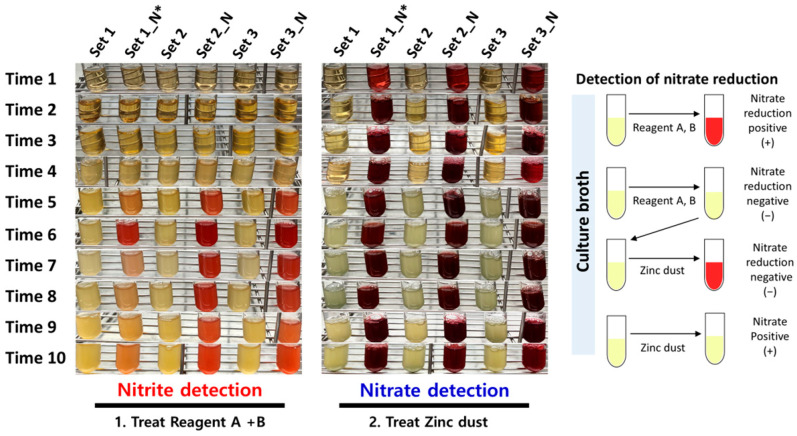
Nitrate reduction test in bacterial co-culture environment. Experimental data for nitrate reduction test to confirm the nitrate-reducing reaction in the co-culture environment for all bacterial species. The left (red color wording indication) is the detection test of nitrite (NO_2_^−^), and the right (blue color wording indication) is the detection test of residual nitrate (KNO_3_) within the co-culture medium. The upper horizontal line in the figure indicates the comparison sets for bacterial co-culture, and the left vertical line represents the cultivation time (* N = Nitrate treatment group).

**Figure 4 microorganisms-13-00279-f004:**
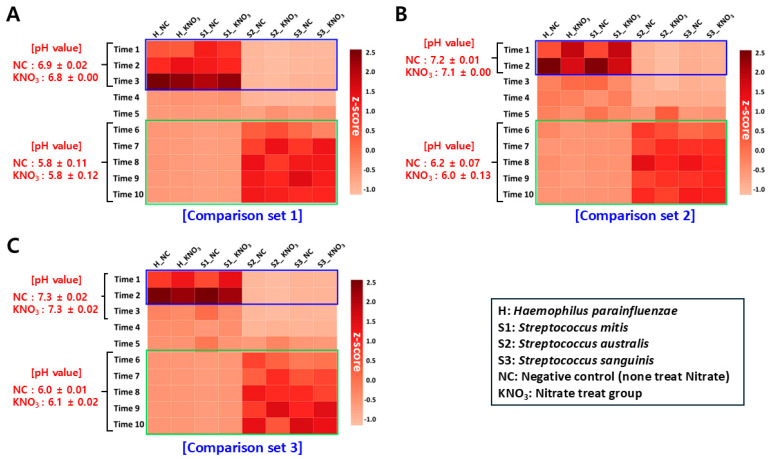
Comparison of relative microbial growth efficiency using qRT-PCR method. Heatmap plots showing the comparison of the relative growth efficiency of each bacterial species within the co-culture environment through the qRT-PCR-based microbial molecular diagnosis method. (**A**–**C**) are separate result data for each comparison set constructed according to the bacterial co-culture experimental conditions. The *x*-axis of the plot indicates each bacterial species in the comparison set according to whether nitrate was added or not, and the *y*-axis indicates the culture medium sampling time. The blue box in the plot indicates when *H. parainfluenzae* has high growth efficiency, and the green box indicates when Streptococci has high growth efficiency.

**Figure 5 microorganisms-13-00279-f005:**
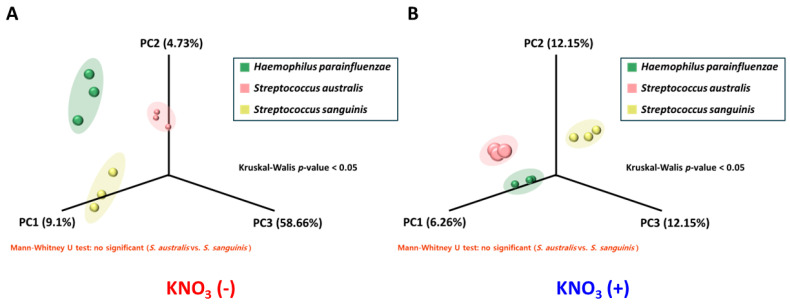
Differences in the metabolite composition between each bacterial species. PCoA (Principal Coordinate Analysis) plots showing the distance dissimilarity in the metabolic composition profiled within the monoculture medium for each bacterial species through the LC/Q-TOF technique. (**A**) is the result data for the no nitrate treatment (nitrate negative) group, and (**B**) is the nitrate treatment (nitrate positive) group.

**Table 1 microorganisms-13-00279-t001:** Overall information on bacterial culture in this study.

Bacterial Culture	Comparison Set	Nitrate Treat ^1^	Culture Media	Bacterial Culture Type	H:S Ratio ^3^
Monoculture	Unity	Negative (−)	BHI ^2^ broth with NAD^+ 2^	Batch	-
		Case (+)			
Co-culture	Set 1	Negative (−)	BHI broth with NAD^+^	Batch	1:3
		Case (+)			
	Set 2	Negative (−)			3: 1
		Case (+)			
	Set 3	Negative (−)		Continuous	
		Case (+)			

^1^ Nitrate treat: KNO_3_ treatment (+: treat, −: none treat); ^2^ BHI: brain–heart infusion media; ^2^ NAD^+^: Nicotinamide Adenine Dinucleotide; ^3^ H:S ratio: *Haemophilus parainfluenzae* and *Streptococcus* sp. Inoculation ratio.

**Table 2 microorganisms-13-00279-t002:** Metabolite screening for each bacterial species based on monoculture.

Comparison Group	Bacterial Species	Raw Compound ^3^	Matched Compound ^4^	Distinct Compound ^5^
Negative group **^1^**	*H. parainfluenzae*	1186	382	229
	*S. australis*	1085	349	22
	*S. sanguinis*	1155	359	20
Case group **^2^**	*H. parainfluenzae*	1043	316	167
	*S. australis*	1065	312	20
	*S. sanguinis*	972	304	17

^1^ Negative group: KNO_3_ (nitrate) non-treated group; ^2^ Case group: KNO_3_ (nitrate) treated group; ^3^ Raw compound: Matching score > 90, *p*-value < 0.05 (only shared), FC > 2 (only shared) MPP fold change analysis data; ^4^ Matched compound: Identification and annotated list of compounds using Agilent MPP software’s ID browser tool; ^5^ Distinct compound: Potential species-specific metabolites detected within each monoculture medium.

**Table 3 microorganisms-13-00279-t003:** Metabolite screening within bacterial co-culture environment.

	Comparison Sets	Groups	Sampling Points	Raw Compound ^5^	Matched Compound ^6^	Distinct Compound ^7^	Lipid Pathway	pH Regulation	Antibacterial Effect
Inter-sampling points ^1^	Set 1	Negative ^3^	1, 4, 10	167	63	-	11	5	14
		Case ^4^		504	130		17	7	33
	Set 2	Negative		59	12		4	2	1
		Case		67	14		3	1	2
	Set 3	Negative		9	1		0	0	0
		Case		22	2		1	0	0
Intra-sampling point ^2^	Set 1	Negative	1	1458	421	19	1	0	6
			4	1328	396	24	6	8	10
			10	1498	428	32	3	4	17
		Case	1	1534	407	39	12	2	9
			4	1232	338	19	5	4	6
			10	1455	386	20	1	1	6
	Set 2	Negative	1	2256	443	28	3	0	6
			4	2210	430	16	3	0	6
			10	2199	436	29	3	1	7
		Case	1	2283	441	26	4	2	6
			4	2219	448	13	0	0	5
			10	2209	454	36	3	0	1
	Set 3	Negative	1	2202	420	17	1	0	8
			4	2144	419	14	1	1	7
			10	2177	431	32	6	0	9
		Case	1	2241	448	12	1	0	5
			4	2333	470	22	3	2	5
			10	2348	461	24	4	0	10

^1^ Inter-sampling points: Metabolite profiling for shared compound between three sampling points; ^2^ Intra-sampling point: Metabolite profiling for distinct compound of each sampling point; ^3^ Negative: KNO_3_ (nitrate) non-treated group; ^4^ Case: KNO_3_ (nitrate) treated group; ^5^ Raw compound: Matching score > 90, *p*-value < 0.05 (only shared), FC > 2 (only shared) MPP fold change analysis data; ^6^ Matched compound: Identification and annotated list of compounds using Agilent MPP software’s ID browser tool; ^7^ Distinct compound: Specific metabolite distinguished by each cultivation time (sampling point).

**Table 4 microorganisms-13-00279-t004:** Metabolic compounds list matched with each monoculture-based metabolite library.

Related Metabolisms	Groups	Matched with *Streptococcus* sp. Specific ^3^ Compounds	Matched with *H. parainfluenzae* Specific Compounds
Antibacterial effect	Negative ^1^	Aurasperone D	3,4,3′,4′-Tetrahydrospirilloxanthin
		Hydroxyprolyl-Tryptophan	Filicin
		8-Acetoxy-4′-methoxypinoresinol 4-glucoside	
		Minocycline	
		3,4,3′,4′-Tetrahydrospirilloxanthin	
	Case ^2^	Dodemorph	Thalassemine
		11-Deoxylandomycinone	5,7-Dihydroxy-3′,4′-dimethoxy-8-(3-hydroxy-3-methylbutyl)-isoflavone 7-glucoside
		Cyclochlorotine	
		Iriomoteolide 1 a	
		Mascaroside	
Lipid pathway	Negative	Leukotriene E3	(±)-Octanoylcarnitine
			(E,E,E)-N-(2-Methylpropyl)hexadeca-2,6,8-trien-10-ynamide
	Case	Palmitoyl Ethanolamide	
		Sulfoglycolithocholate	
		Citranaxanthin	
pH regulation	Negative	Leukotriene E3	
		Traumatic Acid	
	Case		

^1^ Negative: KNO_3_ (nitrate) non-treated group; ^2^ Case: KNO_3_ (nitrate) treated group; ^3^ *Streptococcus* sp. specific: *Streptococcus* sp. include *S. australis*, and *S. sanguinis* specific metabolites screened each monoculture medium.

## Data Availability

The information on metagenome datasets generated during the current study is included in this article, and it is available from the corresponding author upon reasonable request.

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
