# Peer review of "Exploring Competitive Relationship Between Haemophilus parainfluenzae and Mitis Streptococci via Co-Culture-Based Molecular Diagnosis and Metabolomic Assay"

_microorganisms, 2025, doi:10.3390/microorganisms13020279_

Round 1

Reviewer 1 Report

Comments and Suggestions for Authors

The co-culture study focusing on Haemophilus parainfluenzae and mitis streptococci is novel and significant study. Especially the role of nitrate/nitrite metabolism in the interaction is significant as so far not much is known about it's role in modulation the behaviour of the oral kicrobiome and it's effect on oral infectious diseases. The study is well designed, however, there are several areas that need to be addressed:

1. The title needs to spell out the name of Haemophilus. Also, it may be revised to better reflect the content of the manuscript (minor).

2. The abstract mentioned dental caries but those were not studied or well discussed (l. 19-20). Possible extended discussion could be included.

3. l. 77. "various experimental conditions" ... this could be better defined to more precisely address what has been done

4. Why pH was not varied for the monocultures since it appears to play a role? 

5. l. 103. political seed culture? needs to be revised. Few other typos in teh manuscript are also found so the manuscript needs to be carefully proofread 

6. l. 113 - 120. Growth was monitored by OD but that may not exactly reflect growth.. it is unclear why qRT-PCR was not used in addition. Also, plating to ensure the bacteria are live.

7.  l. 161-163. It is not known if the genes selected as probes are regulated under different growth conditions .. that would affect the data 

8. Major: bacterial genomic DNA was isolated and qRT-PCR was performed. This is the major issue in this paper as qRT-PCR is performed in RNA and not on DNA. This needs to be clarified (or revised).

9. Were the studies done in technical or biological replicates (please describe the design)

10. l. 256. 'We consisted" - please revise - unlclear meaning

11. It is impossible to read Panel A in Fig. 1; the font is too small

12. l. 282. wording: "bottom-up" shape ... it is unclear

13. Fig. 1C. The diagram describes genomic DNA isolation followed by qRT-PCR quantification. This needs to be addressed as qPCR would result from such analysis and would account for live and dead bacteria.

14.  Fig. 2. The growth curves are not logarithmic. Effect pf pH  and nitrate/nitrite is not studied here. How many biological replicates were used in this study?

15. Fig. 3. This figure is confusing; if there is nitrite then there should be less nitrate ... Please address in discussion.

16. Haemophilus and strep mitis grow similarly, and different than strep. australis and streo. sanguinus. Please address the differences. 

17. L348-362. Is the pH change also found in monocultures or is this a results of co-culture?

18. Table 4. Negative vs NC is this the same (needs to be addressed in legend). 

19. Traumatic acid is only detected in NC, similarly minoclycline. Thus nitrate treatment affects the metabolome. The viability of the bacteria in the co-culture needs to be determined.

20. l. 603 ..etc, too vaque expression 

21. Fig. 3 and Fig. 4. The time is expressed in hr and in times 1, 2, 3... It would help if that was consistently expressed

Author Response

We sincerely thank you for reviewing our manuscript and providing your valuable comments.
I would appreciate it if you could kindly check the attached PDF file.

Reviewer 2 Report

Comments and Suggestions for Authors

Previous research has indicated that an imbalance between the Streptococci strains and Haemophilus parainfluenzae is linked to dental health problems and that the growth of H. parainfluenzae is inhibited  by mitis Streptococcus produced H2O2 .The current study represents an attempt to understand more deeply the competitive relationship between H. parainfluenzae and mitis Streptococcus, but there are some points that should be taken into consideration while reviewing the manuscript:

Line 19, The goal does not fit the title of the manuscript.

Line 32, the scientific conclusion from the study must be clarified in light of the results.

Line 90, as well as line 245, what is the reason and necessity to mention the results of the previous study (reference 9), especially since it has been published? It would have been efficient to just refer to the study in the introduction part (lines 46-49).

Line 629, the authors stated "we suggest" and then references were given at the end of the sentence!

A more comprehensive discussion of the results of LC/Q-TOF-based metabolite screening is required.

The conclusion part should be rewritten to present the results of the study.

line 668, "Ultimately, our approach and results demonstrate that it is important to maintain a balance between mitis Streptococci and H. parainfluenzae within the human salivary microbiome". In light of the results of the study, how can the balance between streptococci and H. parainfluenzae be maintained within the human salivary microbiome?

Author Response

(The authors gave the same response as above.)

Round 2

Reviewer 2 Report

Comments and Suggestions for Authors

A commendable effort made by the authors in responding to inquiries and responding to the proposed amendments.